# Prevalence of Dengue, Chikungunya and Zika Viruses in Blood Donors in the State of Pará, Northern Brazil: 2018–2020

**DOI:** 10.3390/medicina59010079

**Published:** 2022-12-30

**Authors:** Leticia Martins Lamarão, Angelita Silva Miranda Corrêa, Renata Bezerra Hermes de Castro, Carlos Eduardo de Melo Amaral, Patricia Danin Jordão Monteiro, Mauricio Koury Palmeira, Luane Nascimento Lopes, Angela Neves Oliveira, Maria Salete Maciel de Lima, Caroline Aquino Moreira-Nunes, Rommel Rodríguez Burbano

**Affiliations:** 1Foundation Center for Hemotherapy and Hematology of Pará, Nucleic Acid Test (NAT) Department, Belém 66033-000, PA, Brazil; 2Pharmacogenetics Laboratory, Drug Research and Development Center, Department of Medicine, Federal University of Ceará, Fortaleza 60430-275, CE, Brazil; 3Oncology Research Center, Department of Biological Sciences, Federal University of Pará, Belém 66073-005, PA, Brazil; 4Molecular Biology Laboratory, Ophir Loyola Hospital, Belém 66063-240, PA, Brazil; 5Human Cytogenetics Laboratory, Biological Science Institute, Federal University of Pará, Belém 66075-110, PA, Brazil

**Keywords:** Chikungunya virus, dengue virus, Zika virus, blood donors, transfusion

## Abstract

Arboviruses have been reported over the years as constant threats to blood transfusion recipients, given the high occurrence of asymptomatic cases and the fact that the presence of viremia precedes the onset of symptoms, making it possible that infected blood from donors act as a source of dissemination. This work aims to identify the prevalence of dengue virus (DENV), Zika virus (ZIKV) and Chikungunya virus (CHIKV) infection in blood donors during epidemic and non-epidemic periods; classify the donor as symptomatic or asymptomatic; and verify the need to include DENV, CHIKV and ZIKV in the nucleic acid test (NAT) platform in northern Brazil. We investigated 36,133 thousand donations in two years of collection in Northern Brazil. One donor was positive for DENV and one for CHIKV (0.002% prevalence). As the prevalence for arboviruses was low in this study, it would not justify the individual screening of samples from donors in a blood bank. Thus, DENV- and CHIKV-positive samples were simulated in different amounts of sample pools, and both were safely detected by molecular biology even in a pool of 14 samples, which would meet the need to include these three viruses in the routine of blood centers in endemic countries such as Brazil.

## 1. Introduction

The global increase in re-emerging pathogens such as dengue virus (DENV), Zika virus (ZIKV) and Chikungunya virus (CHIKV) poses a threat to public health and the safety of the blood supply. All three pathogens are arthropod-borne viruses (arboviruses), primarily Aedes aegypti [1]. The female mosquito is contaminated when it feeds on the blood of an infected person during the initial febrile period (viremia); the virus lodges in the cells that line the intestine and then spreads to other tissues of the insect, such as the salivary gland, where it stays until the moment of transmission [2,3].

The viruses seem to have no adverse effect on the insect, which remains infected for life without showing symptoms [4]. The transmission of DENV, and currently of ZIKV and CHIKV, is of great concern to public health worldwide, as these arthropods are widely distributed in tropical, subtropical (*Aedes aegypti*) and temperate (*Aedes albopictus*) regions, covering a huge contingent of susceptible individuals [5,6]. The rapid spread of these three viruses and their epidemic potential are worrying, especially in territories with the circulation of other arboviruses, due to the difficulty in differential diagnosis and the overload of health services. Control measures are the same as those recommended for dengue, based on health education and vector control [7,8].

From January 2008 to October 2011, the estimated number of travelers exposed to CHIKV and DENV arriving in Italy was higher compared to reported cases, suggesting a possible underestimation of the actual number of imported cases, and reinforcing the need to analyze the prevalence of these arboviruses [9].

There are few data in the world literature on the prevalence of DENV, ZIKV and CHIKV viruses in blood donors and their clinical consequences in recipients of blood contaminated by transfusion. Arboviruses are constant threats to the human population, including recipients of blood transfusion, as well as through organ donation, given the high occurrence of asymptomatic cases and the fact that the presence of viremia precedes the onset of symptoms, allowing infected blood from donors to act as a source of dissemination [3,10,11,12]. In Brazil, the Ministry of Health uses the nucleic acid test (NAT) as a tool to screen out donors for human immunodeficiency virus (HIV), hepatitis C (HCV) and hepatitis B (HBV), thereby reducing the risk of disease transmission by an early detection of viral antigens [13,14].

Epidemiological surveillance in the blood bank can be essential to quickly identify infected donors before a possible transfusion. The aim of this study is to contribute to an analysis of the risk of transfusion of arboviruses through the donation of contaminated blood and with a pooled analysis methodology for DENV, CHIKV and ZIKV that safely allows agility in assessing the quality of blood components. In this way, if any pool was positive for one or more of these three arboviruses, the pool would be separated and the samples that make up the pool would be tested individually and thus it would be relatively simple to identify the donor or positive donors for DENV, CHIKV and ZIKV.

## 2. Materials and Methods

### 2.1. Ethical Statement

A total of 36,133 donations collected at the Foundation of Hematology and Hemotherapy of the State of Pará (HEMOPA), located in the city of Belém, which supports blood transfusion in the North Region of Brazil, were investigated. This is a fully representative sample, as all blood samples from donors sent for molecular screening of the HCB HBV and HIV viruses at HEMOPA.

HEMOPA foundation is the only reference center in study region, the state of Pará, Brazil, which in 2018 had 8.51 million inhabitants, distributed over 1,247,960 km^2^, according to the population estimate carried out by the Brazilian Institute of Geography and Statistics (IBGE).

In the period from 1 June 2018 to 30 June 2020, donors referred for collection of whole blood bags or platelet apheresis were included in this research. This research was carried out in accordance with the guidelines and standards of the Declaration of Helsinki and was approved by the Ethics Committee for Research with Human Beings of the HEMOPA Foundation (CAAE 43610615.7.0000.5174) and the donors signed the Free and Informed Consent Term at the time of clinical screening, authorizing their participation in the research.

### 2.2. Collection and Processing of Samples

The samples were processed for automated extraction and purification of nucleic acids according to NAT’s laboratory routine at the HEMOPA Foundation [15]. Genetic material was extracted from each donor individually and DENV (DENV 1, DENV 2, DENV 3 and DENV 4), ZIKV and CHIK viruses were detected using one step reverse transcription polymerase chain reaction (RT-qPCR) technique in a multiplex assay, using primers and Taqman^®^ probes (ThermoFisher Scientific, Waltham, MA, USA) on the Applied Biosystems 7500 Real-Time PCR System (ThermoFisher Scientific, Waltham, MA, USA).

To assess the sensitivity of the methodology, we also performed an assay where we extracted RNA from samples in pools of 6, 8, 10, 12, 14, 16, 18, and 20 donors [16]. Descriptive measurements were performed calculating the prevalence of DENV and ZIKV.

## 3. Results

One positive donor for CHIKV and one for DENV were detected out of 36,133 samples investigated. The total prevalence identified during the entire study period detecting arboviruses in blood donors was 0.005%. The sample infected by DENV was found during a non-epidemic period and had a prevalence of 0.002%. The prevalence of 0.002% is repeated when analyzing CHIKV but found in an epidemic period. As for ZIKV, in this study we did not detect any positive case (prevalence 0.00%).

Positive donors for DENV and CHIKV were contacted by our project team and after clinical evaluation, they were classified as symptomatic after donation; however, both had mild symptoms such as mild fever, malaise, body aches, and tiredness, and both did not seek medical attention at the time of donation. In this research, blood donations positive for DENV and CHIKV had no recipients.

DENV and CHIKV positive samples were simulated in different amounts of sample pools, up to 20 samples per pool (Table 1). It was observed that if the DENV positive sample had been performed within a pool of 14 different donor plasmas, it would have been automatically detected in the system protocol. As for CHIKV, it was observed that even if this sample had been composed of a pool of 20 plasma samples, it would have been automatically detected by the system protocol. If it were necessary to standardize detection for DENV and CHIKV in a pool of samples, it would be prudent and safe to use a pool of up to 14 different plasmas for the two viruses.

## 4. Discussion

In the study region, the first semester coincides with the rainy season, and is considered seasonally epidemic for arboviruses and contributes significantly to the proliferation and development of Aedes aegypti [17]. We identified the prevalence of infections by the arboviruses including DENV, CHIKV and ZIKA in blood donors during epidemic and non-epidemic periods, the donors being classified as symptomatic or asymptomatic. Additionally, we verified the need to include screening for this virus in the blood centers and the possibility of this screening being performed in a pool of samples.

Due to the COVID-19 pandemic, the focus of research on arboviruses lost strength; for this reason, the present study intends to update DENV, ZIK and CHIKV in blood donors before the pandemic that began in Brazil in the first half of 2020. In the current study, one donor was positive for CHIKV and one for DENV (a prevalence of 0.002% for each arbovirus). This result showed a much lower prevalence in relation to HIV (0.06%), HBV (0.02%) and HCV (0.01%), found by the HEMOPA Foundation in the same study samples. CHIKV was found in an epidemic period and DENV in a non-epidemic period. This prevalence is much lower than that of other agents in the same region and study period, such as human immunodeficiency virus (HIV) 0.06%, hepatitis B virus (HBV) 0.02% and hepatitis C virus (HCV) 0.01%. Both DENV- and CHIKV-positive donors were classified as having few symptoms after donation. The risk of transmission and severity were not evaluated, as blood from donors positive for DENV and CHIV did not have recipient patients.

The sample infected by DENV was found in a non-epidemic period and had a prevalence of 0.002%, which suggests that in non-epidemic periods, the chances of blood donors being able to transmit the DENV virus through blood donation may increase. The prevalence of DENV in blood donors varies according to the location and the epidemic and non-epidemic period. The prevalence in these locations ranged from 0.04 to 0.9% [2,18,19,20,21,22,23,24]. In the case of DENV, it is known that transmission by blood transfusion in recipients aggravates the disease [1,24].

The prevalence of 0.002% is repeated when analyzing CHIKV, but found in an epidemic period, and is lower than those found in other studies. In 2009, during an epidemic in a province of Thailand, a CHIKV prevalence of 0.03 to 2.97% was found in blood donors [25]. Similarly, in 2014 in Puerto Rico, during an epidemic period, the prevalence of the virus was identified in 2.1% of blood donors [26]. So far, in the world literature, it has not been possible to predict the transfusion transmission of this virus.

In August 2009, a group from the Association for the Advancement of Blood and Biotherapies (a non-profit international association representing individuals and institutions involved in the fields of transfusion medicine and biotherapies) reviewed emerging infectious disease agents that pose a real or theoretical threat to transfusion safety. One of the agents that received the highest priority was the dengue virus. Although the focus of the study was on the United States and Canada, the threat of DENV applies worldwide [27]. The detection of a positive case of DENV was found in the first half of 2019, a period of greater circulation of the disease vectors, and a period in which almost all cases occur in Brazil [28,29], suggesting the need for attention to be paid to this agent.

As for ZIKV, in this study we did not detect any positive case (prevalence 0.00%). A systematic review and meta-analysis found a prevalence of 0.85% of the virus in blood donors, and this prevalence varied according to epidemic and non-epidemic periods and localities [30]. In Campinas, a city located in southeastern Brazil, in the years 2015 and 2016, a prevalence of ZIKV was observed that ranged from 0.05 to 0.17% in blood donors [23,30]. In Colombia, another South American country, the prevalence of ZIKV was on the order of 2.62% among blood donors in the period from 2015 to 2016 [31]. The fact that we did not find cases in our study reinforces that since 2018, clinical screening and population awareness have probably allowed this prevalence to decrease among blood-bank donors [32].

The development of the apparent symptoms of the disease caused by ZIKV in blood-component recipients has not yet been proven [33,34,35]. As far as we know, there are no descriptions of prevalence in the northern region of Brazil and, in this study, we did not detect this virus in blood donors. Positive donors for DENV and CHIKV were contacted by the project team and after clinical evaluation were classified as symptomatic after donation; however, both had mild symptoms such as mild fever, malaise, body aches, and tiredness, and both did not seek medical attention at the time of donation. In this research, blood donations positive for DENV and CHIKV had no recipients.

DENV and CHIKV positive samples were simulated in different amounts of sample pools, with up to 14 samples per pool. It was observed that if the DENV-positive sample had been performed within a pool of 14 different donor plasmas, it would have been automatically detected in the system protocol. As for CHIKV, it was observed that even if this sample had been composed of a pool of 20 plasma samples, it would have been automatically detected by the system protocol. If it were necessary to standardize detection for DENV and CHIKV in a pool of samples, it would be prudent and safe to use a pool of up to 14 different plasmas for the two viruses.

The need for integrated human and entomological surveillance to monitor the spread of vector-borne diseases and to implement public health measures to prevent transmission and control of these diseases in humans has been demonstrated in imported Chikungunya and dengue infections in Italy [9]. The present study contributed to the assessment of DENV, CHIKV and ZIKV transfusion-transmission risk in a blood bank in an extensive tropical region, represented by the state of Pará in northern Brazil, and estimated the number of pooled samples that can be analyzed by the NAT tool that is standardized for the evaluation of other bloodborne viruses.

## 5. Conclusions

In conclusion, this study indicates the prevalence of DENV and CHIKV in blood donors of northern Brazil. Based on results of the study, we concluded the need to include continuous and mandatory molecular screening tests for DENV, CHIKV and ZIKV in blood centers, in order to prevent these agents from being transmitted to recipients of blood components by asymptomatic patients. In case it is necessary to standardize the detection of these arboviruses in a sample pool, it would be safe to use a pool of up to 14 different plasmas for DENV and CHIKV.

## Figures and Tables

**Table 1 medicina-59-00079-t001:** Identification and detection values of the positive sample for DENV and CHIKV in different quantitative sample pools.

Average of Detection Values								
Number of Pooled Samples	6	8	10	12	14	16	18	20
Dengue	23.7	24.2	25.4	28.8	28.5	ND *	ND *	ND *
Chikungunya	22.8	23.1	23.8	24.2	24.1	24.4	24.5	25.0

ND: not detected; * Presence of curve below detection limit.

## Data Availability

All experimental data and analysis results were stored in the NAT laboratory computer and all samples were stored in the HEMOPA Blood Bank, which are available to be reviewed. No public database is available to deposit our data. The datasets used and/or analyzed during the current study are available from the corresponding author on reasonable request.

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
