# Peer review of "Prevalence of Dengue, Chikungunya and Zika Viruses in Blood Donors in the State of Pará, Northern Brazil: 2018–2020"

_medicina, 2022, doi:10.3390/medicina59010079_

Round 1

Reviewer 1 Report

In the pre-pandemic era, communicable diseases mortality has reduced in high-income countries, although infectious, such as arboviruses, due to their lack of symptoms, frequently contribute to morbidity, especially due to blood donations. In this context, aim of the study under review is identified the prevalence of infections by the arboviruses Dengue, Chikungunya and ZIKA in blood donors during epidemic and non-epidemic periods.

The subject under study is certainly important. The article presents some interesting results, but it must be improved before publication especially related to some methodological issue. At this stage it seems a local epidemiological report more than a scientific paper.

Title: it should be improved, reporting the precise geographical area and the time of the study.

Abstract: it is information about the article, its content, goals, objectives, assumptions, method used, results and conclusions. In this case the main objective must be clarified, the method used is presented at an insufficient level, and the conclusions in your case are presented in a cursory manner.

Introduction: the authors should make it clear about what is the gap in the international literature before introducing the problem of Australia. Moreover, the general problem of arboviruses including the risk of the possible importation of these infectious disease (refer to article with doi: 10.1111/j.1708-8305.2012.00640.x.). Finally, better specify what is the contribution of the study to the literature? What are the possible implications of the study?

Methods: The study was conducted using isolates from 2018-2020, I understand that the Covid-19 pandemic may have focused the researches on it, but the epidemiological data the author are presenting is old. The enrolment procedure must be better specified. How did the authors choose the way to select the samples? This can represent a great bias origin. How did they avoid the selection bias? The authors do not propose a minimum sample size, what is the reference population of the enrolled hospital? How large is the reference population? Without the numerical identification of the reference population is not clear the validity of the study. A non-representative sample is by its self a non-sense-study.

Discussion: I also suggest expanding. Emphasize the contribution of the study to the literature, the implications and recommendations based on the conclusions. The issue of the risk of importation must be discussed and referenced (see the above mentioned reference). Limits section can also be improved.

Author Response

Dear reviewer, my co-authors and I would like to thank you for the suggestions made during this high-quality review and then we present the answers to the questions.

We inform that with the reviews and suggestions, we were able to improve the idea presented by our work and we appreciate the opportunity. We hope this review has left the article suitable for publication in this high-impact journal and respect in the area.

Kind Regards.

Response to reviewer 1

In the pre-pandemic era, communicable diseases mortality has reduced in high-income countries, although infectious, such as arboviruses, due to their lack of symptoms, frequently contribute to morbidity, especially due to blood donations. In this context, aim of the study under review is identified the prevalence of infections by the arboviruses Dengue, Chikungunya and ZIKA in blood donors during epidemic and non-epidemic periods.

The subject under study is certainly important. The article presents some interesting results, but it must be improved before publication especially related to some methodological issue. At this stage it seems a local epidemiological report more than a scientific paper.

Response: Suggested improvements have been made.

Title: it should be improved, reporting the precise geographical area and the time of the study.

Response: Suggested improvements have been made.

Abstract: it is information about the article, its content, goals, objectives, assumptions, method used, results and conclusions. In this case the main objective must be clarified, the method used is presented at an insufficient level, and the conclusions in your case are presented in a cursory manner.

Response: Suggested improvements have been made.

Introduction: the authors should make it clear about what is the gap in the international literature before introducing the problem of Australia. Moreover, the general problem of arboviruses including the risk of the possible importation of these infectious disease (refer to article with doi: 10.1111/j.1708-8305.2012.00640.x.). Finally, better specify what is the contribution of the study to the literature? What are the possible implications of the study?

Response: Suggested improvements have been made.

Methods: The study was conducted using isolates from 2018-2020, I understand that the Covid-19 pandemic may have focused the researches on it, but the epidemiological data the author are presenting is old. 

Response: The reviewer is right due to the Covid pandemic, the focus of research on arboviruses lost strength, for this reason we were able to update the DENV, ZIK and CHIV data before the pandemic that began in Brazil in the first half of 2020, we justify this in the discussion

The enrolment procedure must be better specified. How did the authors choose the way to select the samples?

This can represent a great bias origin. How did they avoid the selection bias? The authors do not propose a minimum sample size, what is the reference population of the enrolled hospital? How large is the reference population? Without the numerical identification of the reference population is not clear the validity of the study. A non-representative sample is by itself a non-sense-study.

Response: The reviewer is right, this information was missing This is a fully representative sample, since all blood samples from donors sent for molecular screening of the HCB HBV and HIV viruses from HEMOPA were used for the DENV, CHIKV and ZIKV research, which it is the only reference center in the study region. the state of Pará, Brazil, which in 2018 had 8.51 million inhabitants, spread over 1,247,960 km2, according to the population estimate carried out by the Brazilian Institute of Geography and Statistics (IBGE). This was introduced in the material and methods section.

Discussion: I also suggest expanding. Emphasize the contribution of the study to the literature, the implications and recommendations based on the conclusions. The issue of the risk of importation must be discussed and referenced (see the above-mentioned reference). Limits section can also be improved.

Response: Suggested improvements have been made.

Reviewer 2 Report

This brief report entitled “Prevalence of Dengue virus, Chikungunya and Zika infection in blood donors of northern of Brazil” is novel and interesting one. It emphasizes on an important issue related to human health and well being.

I suggest a slight modification in the Title:
Proposed Title: Prevalence of Dengue, Chikungunya and Zika viruses in blood donors of Northern Brazil

I have gone through the manuscript and found the following observations:

Abstract: The abstract is poorly written. It needs to be succinct and precise. It is preferable to present extraneous details in the discussion chapter rather than in the abstract.

Introduction: There are numerous grammatical and typographical errors in the Introduction. There is no chronological order to the information in this chapter. Similar to this, the paragraphs are too brief, which shows a weak presentation of the information. Since there is no line numbering in the pdf file, it is very challenging to address each one individually. I have therefore suggested a few changes to the pdf file that is attached. Overall, the authors need a second supervisor to help with the technical write-up.

Conclusions: These are more like recommendations rather than conclusion. Conclusion should be drawn based on the actual findings of the study. 

Note: In the attached pdf file, changes are highlighted in red throughout the manuscript.

Author Response

Dear reviewer, my co-authors and I would like to thank you for the suggestions made during this high-quality review and then we present the answers to the questions.

We inform that with the reviews and suggestions, we were able to improve the idea presented by our work and we appreciate the opportunity. We hope this review has left the article suitable for publication in this high-impact journal and respect in the area.

Kind Regards.

Reviewer 2

This brief report entitled “Prevalence of Dengue virus, Chikungunya and Zika infection in blood donors of northern of Brazil” is novel and interesting one. It emphasizes on an important issue related to human health and well-being.

I suggest a slight modification in the Title:
Proposed Title: Prevalence of Dengue, Chikungunya and Zika viruses in blood donors of Northern Brazil

Response: Suggested improvements have been made.

I have gone through the manuscript and found the following observations: 

Abstract: The abstract is poorly written. It needs to be succinct and precise. It is preferable to present extraneous details in the discussion chapter rather than in the abstract.

Response: Suggested improvements have been made.

Introduction: There are numerous grammatical and typographical errors in the Introduction. There is no chronological order to the information in this chapter. Similar to this, the paragraphs are too brief, which shows a weak presentation of the information. Since there is no line numbering in the pdf file, it is very challenging to address each one individually. I have therefore suggested a few changes to the pdf file that is attached. Overall, the authors need a second supervisor to help with the technical write-up.

Response: Suggested improvements have been made.

Conclusions: These are more like recommendations rather than conclusion. Conclusion should be drawn based on the actual findings of the study. 

Response: Suggested improvements have been made.

Round 2

Reviewer 1 Report

The paper was improved following my suggestions and it is now suitable for publication

Author Response

Dear reviewer, my co-authors and I would like to thank you for the suggestions made during this high-quality review and then we present the answers to the questions.

We inform that with the reviews and suggestions, we were able to improve the idea presented by our work and we appreciate the opportunity. We hope this review has left the article suitable for publication in this high-impact journal and respect in the area.

Kind Regards.

Response to reviewer 1

Dear Revisor, suggested improvements and check of all minor spelling errors have been made.

Reviewer 2 Report

I suggest a slight modification in the Title:

Proposed title: Prevalence of Dengue, Chikungunya and Zika viruses in blood donors in the State of Pará, Northern Brazil

or If its mandatory to mention the period, then the proposed title could be:

Prevalence of Dengue, Chikungunya and Zika viruses in blood donors in the State of Pará, Northern Brazil: 2018-2020

In the hard copy, my small edits are highlighted in red and have been suggested changes to the manuscript (attached herewith)

Author Response

Dear reviewer, my co-authors and I would like to thank you for the suggestions made during this high-quality review and then we present the answers to the questions.

We inform that with the reviews and suggestions, we were able to improve the idea presented by our work and we appreciate the opportunity. We hope this review has left the article suitable for publication in this high-impact journal and respect in the area.

Kind Regards.

Reviewer 2

I suggest a slight modification in the Title:

Proposed title: Prevalence of Dengue, Chikungunya and Zika viruses in blood donors in the State of Pará, Northern Brazil

or If its mandatory to mention the period, then the proposed title could be:

Prevalence of Dengue, Chikungunya and Zika viruses in blood donors in the State of Pará, Northern Brazil: 2018-2020

R= We really appreciate your suggestion, and we chose the second option.

In the hard copy, my small edits are highlighted in red and have been suggested changes to the manuscript (attached herewith)

R = All the suggested modifications were also made.